

# An examination of clinical differences between carriers and non-carriers of chromosome 8q24 risk alleles in a New Zealand Caucasian population with prostate cancer

Karen S. Bishop[1], Dug Yeo Han[2,3], Nishi Karunasinghe[1], Megan Goudie[4], Jonathan G. Masters[4] and Lynnette R. Ferguson[1,2,3]

[1] Auckland Cancer Society Research Centre, Faculty of Medical and Health Sciences, University of Auckland, Auckland, New Zealand
[2] Nutrigenomics New Zealand, University of Auckland, Auckland, New Zealand
[3] Discipline of Nutrition and Dietetics, Faculty of Medical and Health Sciences, University of Auckland, Auckland, New Zealand
[4] Urology Department, Auckland District Health Board, Auckland, New Zealand

## ABSTRACT

**Background.** Prostate cancer makes up approximately 15% of all cancers diagnosed in men in developed nations and approximately 4% of cases in developing nations. Although it is clear that prostate cancer has a genetic component and single nucleotide polymorphisms (SNPs) can contribute to prostate cancer risk, detecting associations is difficult in multi-factorial diseases, as environmental and lifestyle factors also play a role. In this study, specific clinical characteristics, environmental factors and genetic risk factors were assessed for interaction with prostate cancer.

**Methods.** A total of 489 prostate cancer cases and 427 healthy controls were genotyped for SNPs found on chromosome 8q24 and a genetic risk score was calculated. In addition the SNPs were tested for an association with a number of clinical and environmental factors.

**Results.** Age and tobacco use were positively associated, whilst alcohol consumption was negatively associated with prostate cancer risk. The following SNPs found on chromosome 8q24 were statistically significantly associated with prostate cancer: rs10086908, rs16901979; rs1447295 and rs4242382. No association between Gleason score and smoking status, or between Gleason score and genotype were detected.

**Conclusion.** A genetic risk score was calculated based on the 15 SNPs tested and found to be significantly associated with prostate cancer risk. Smoking significantly contributed to the risk of developing prostate cancer, and this risk was further increased by the presence of four SNPs in the 8q24 chromosomal region.

Corresponding author
Karen S. Bishop,
k.bishop@auckland.ac.nz

## INTRODUCTION

Population differences in cancer incidence may reflect differences in genotype and intake of, or exposure to cancer promoting or preventative factors. Identification of genetic risk factors for cancer often utilises linkage studies in high-risk families, but this approach has proved difficult when applied to prostate cancer due to interacting lifestyle and environmental factors (*Crawford, 2003*; *Xu, Sun & Zheng, 2013*). The only clearly established risk factors for prostate cancer are increasing age, family history of prostate cancer and ethnicity (*Crawford, 2003*; *Nordström et al., 2013*; *Sun et al., 2011*), although adoption of a Western diet, increased life expectancy and PSA testing are thought to contribute to the rise in detection of prostate cancer in Asian countries (*Xu, Sun & Zheng, 2013*).

Although there is evidence for a significant hereditary component of prostate cancer susceptibility, results are inconsistently replicated. A positive family history of prostate cancer is associated with increased risk, but relative risk (RR) ratios vary from one study to the next (*Goh et al., 2012*). Twin studies have been used to reveal the heritability of prostate cancer and values have ranged from 0.36 to 0.57 (*Ahlbom et al., 1997*; *Lichtenstein et al., 2000*; *Neale et al., 2005*; *Page et al., 1997*) leaving no doubt regarding a strong genetic component. In addition, in a review of genetic association studies, Xu et al. (*Goh et al., 2012*; *Xu, Sun & Zheng, 2013*) concluded that RR of prostate cancer was highest if a primary relative, particularly a brother, was diagnosed with prostate cancer before the age of 60 years.

In numerous studies the association between single nucleotide polymorphisms (SNPs) in the 8q24 region and prostate cancer, have been reported (*Amundadottir et al., 2006*; *Gudmundsson et al., 2007*; *Wang et al., 2011*). For example the A allele of SNP rs1447295 was associated with an increased prostate cancer risk in Japanese, Native, Latino, African and European Americans (*Freedman et al., 2006*; *Gudmundsson et al., 2007*; *Severi et al., 2007*). However, reports on this SNP explain only a portion of the signal described (*Amundadottir et al., 2006*), and fail to account for all of the heritability. Of the five haplotype blocks (on 8q24) associated with various cancers, three solely and independently contribute to prostate cancer risk (*Ghoussaini et al., 2008*).

Clinical characteristics play an important role in the selection of appropriate treatment and prognosis for prostate cancer (*Heidenreich et al., 2014*). However, it is likely that clinical characteristics interact with environmental and genetic risk factors to determine risk of prostate cancer progression. The environmental factors that have been identified as playing a possible role in the development or progression of prostate cancer include various aspects of dietary intake, alcohol consumption, tobacco use, physical activity levels and energy balance, although evidence is inconsistent (*Wolk, 2009*; *Xu, Sun & Zheng, 2013*). However, it is difficult to clearly delineate risk due to the influence of confounding factors as it is difficult or impossible to control the effect of numerous variables and thus tease out the impact of individual factors. However, an individual's genes, physiological state and environmental exposures must be considered when assessing disease risk and recommending treatment and lifestyle interventions (*Davis & Milner, 2007*).

SNPs found in 8q24 regions 1-3 and thought to be associated with prostate cancer risk will be examined in this study and tested for interactions with clinical and environmental factors in a New Zealand population.

## MATERIALS AND METHODS

A retrospective case-control, population-based study was performed to explore the association between 8q24 genotypes and the clinical manifestations of prostate cancer.

A total of 916 male participants (489 with prostate cancer and 427 healthy controls) were recruited from Auckland, New Zealand (NZ) between 2006 and 2014. The study was more than adequately powered based on calculations carried out prior to recruitment on expected *GPX1* rs1050450T allele frequency (24 volunteers were required for each group based on a difference in allele frequency of 0.058, significance = 0.05% and power = 80%). Only those who self-reported as having European ancestry were included in this analysis. Due to very small numbers, and therefore low statistical power, it was decided not to analyse data from other ethnic groups. Control participants were a group with no known cancers at recruitment (with the exception of skin cancers). The volunteers were recruited with written informed consent and formed part of either the Urology Study or the Selenium Supplementation Study carried out by the Auckland Cancer Society Research Centre, University of Auckland after receiving approval of the study from the Health and Disabilities Ethics Committees: Northern Y Regional Ethics Committee, NZ, (Ethics Ref: NTY/05/06/037 and NTY/06/07/060). Men with prostate cancer who met the study requirements were selected from the databases from the Auckland Regional Urology facility. This database incorporated the Auckland District Health Board, Counties Manukau District Health Board, Waitemata District Health Board and private practices within the Auckland region. Hardcopy invitations were sent to all eligible men with an approximately 25% response rate. Among the Selenium Supplementation Study participants, only those who consented to the use of their blood samples for future NZ Ethics Committee accredited studies were used in this analysis. The following data were collected and analysed: date of birth, height, weight, chronic medication, smoking status, family history of cancer, diagnostic PSA and Gleason score (where relevant). Data were collected at the time of enrolment. For the prostate cancer group, enrolment initially took place within 12 months of diagnosis (60.2% of those enrolled) and thereafter enrolment criteria were modified such that diagnosis could have taken place at any time prior to enrolment. Gleason scores were obtained from biopsy histopathology unless participants underwent a prostatectomy, in which case the post-surgical Gleason score was used. Alcohol consumption was categorised as "Yes" or "No" such that "Yes" applied to anyone who consumed one or more alcoholic units per week over the past year (prior to recruitment). Smoking was categorised as "never" or "ever" smoked. "Ever" smoked presented both "present" and "former" smokers, as there was insufficient power to analyse "present" smokers on their own. The study investigators were not blinded to group allocation.

Blood collection and processing: Blood samples were collected into EDTA vaccutainer tubes (Becton Dickinson, Plymouth, England) from study participants at enrolment. Total

**Table 1 Association of SNPs with prostate cancer-from the literature.** Association of 15 prostate cancer susceptibility single nucleotide polymorphisms, found on 8q24, with the weighted genetic risk score and assigned weights.

| SNP | Tested Allele | Published OR | Weight LN (OR) | % of total weight | Source |
|---|---|---|---|---|---|
| rs10086908 | T | 1.25 from stage 1 | 0.223 | 5.94% | *Al Olama et al. (2009b)* |
| rs1016342 | C | 1.26 | 0.231 | 6.15% | *Pal et al. (2009)* |
| rs1016343 | T | 1.11 | 0.104 | 2.77% | *Al Olama et al. (2009b)* |
| rs1378897 | A | 1.26 (ns) | 0.231 | 6.15% | *Pal et al. (2009)* |
| rs1447295 | A | 1.58 | 0.457 | 12.17% | *Gudmundsson et al. (2009)* |
| rs16901979 | A | 1.8 | 0.588 | 15.65% | *Gudmundsson et al. (2009)* |
| rs16902094 | G | 1.21 | 0.191 | 5.09% | *Gudmundsson et al. (2009)* |
| rs4242382 | A | 1.39 | 0.329 | 8.76% | *Teerlink et al. (2014)* |
| rs445114 | T | 1.14 | 0.131 | 3.49% | *Gudmundsson et al. (2009)* |
| rs620861 | C | 1.11 from stage 1 (ns) | 0.104 | 2.77% | *Al Olama et al. (2009b)* |
| rs6470494 | T | 1.00[a] | 0.0 | 0.0% | *Pal et al. (2009)* |
| rs6470517 | A | 1.58 | 0.457 | 12.17% | *Pal et al. (2009)* |
| rs6983267 | G | 1.19 | 0.174 | 4.63% | *Al Olama et al. (2009b)* |
| rs7000448 | T | 1.23 | 0.207 | 5.51% | *Ghoussaini et al. (2008)* |
| rs871135 | G | 1.39 | 0.329 | 8.76% | *Pal et al. (2009)* |

**Notes.**

[a]OR from *Pal et al. (2009)* was 1.00. In *Liu, Wang & Han (2011)* a meta-analysis was carried out with the following results: GWAS meta-analysis OR = 1.14; Replication meta-analysis OR = 1.00; All meta-analyses OR = 1.14. It was decided to use the OR value from the Replication meta-analysis.

SNP, Single nucleotide polymorphism; ns, Not significant; OR, Odds ratio; LN, Natural log.

genomic DNA extraction was carried out with the QIAamp DNA Blood Mini Kit (Qiagen, Hilden, Germany). A fully automated procedure on the QIAcube (Qiagen) was followed according to the manufacturer's recommendations. DNA was quantified and checked for purity using a Nanodrop ND1000 (Thermo Fisher Scientific, Waltham, MA, USA), stored at −20 °C and used for genotyping at a concentration of 10 ng/μl.

Genotyping: SNPs were selected based on results from a literature search targeting SNPs that are associated with prostate cancer and found on chromosome 8q24 (*Al Olama et al., 2009b*; *Ghoussaini et al., 2008*; *Gudmundsson et al., 2009*; *Gudmundsson et al., 2007*; *Pal et al., 2009*). None of the SNPs at the time of selection were known to be in LD with each other, and SNPs subsequently found to be in LD were removed from the analysis. Genotyping was performed using a Sequenom MassArray system (Sequenom, San Diego, CA, USA). The products were spotted and fired and run on a MALDI-TOF mass spectrometer. The quality of the genotyping was assessed by checking the results from CEU HapMap samples ($n = 6$) and comparing them with SNP data available on the HapMap Genome Browser release #28, build #37.3 (http://hapmap.ncbi.nlm.nih.gov), as well as by comparing results from duplicate samples. SNP data are deposited in the Figshare repository figshare.com/s/750929b0829d11e583c306ec4bbcf141.

Genetic Risk Score (GRS): A GRS was calculated in order to evaluate the accumulative effects of all the 8q24 SNPs tested. A weighted GRS was created by using 15 SNPs assessed in this study (Table 1) (*De Jager et al., 2009*). Firstly, the weight value of each SNP was calculated by using the natural log of the published odds ratios and assigning the weight

**Table 2   Lifestyle and clinical characteristics of the study participants.**

| Phenotypic variables | Status | Malignant[a] (N = 489) | Control[a] (N = 427) | OR (95% CI) | P value |
|---|---|---|---|---|---|
| Smoking status: N (%) | Ever smoked (Current/Former) | 227 (64.5) | 144 (37.0) | 2.28 (1.62–3.21) | 2.32e−06 |
| | Never | 125 (35.5) | 245 (63.0) | 1.00 | |
| | | 137 | 38 | | |
| Alcohol: N (%) | Yes | 265 (75.3) | 338 (86.4) | 0.57 (0.37-0.90) | 0.0147 |
| | No | 87 (24.7) | 53 (13.6) | 1.00 | |
| Missing Data[b] | | 137 | 36 | | |
| Age at Diagnosis: Mean (SD) | | 65.8 (8.2) | 53.1 (13.6) | 1.10 (1.08–1.12) | 9.88e−27 |
| BMI: Mean (SD) | | 27.3 (3.8) | 26.8 (3.7) | 1.02 (0.98–1.07) | 0.2934 |

Notes.

[a] N, Sample size (complete data set for all four variables with no missing data); OR, Odds ratio; CI, Confidence interval; SD, Standard deviation; BMI, Body mass index; "Yes", 1 or more alcoholic units a week;

[b] Missing Data, At least one missing datum point amongst the four variables.

value of each SNP made up to a total of 100%. Secondly, the weighted GRS (named wGRS15 in this study) was calculated by multiplying the weighted value of the SNP by the number of the tested allele of the SNP for each participant. Thirdly, these values were then summed across all of the 15 SNPs (*De Jager et al., 2009*).

Statistical Analysis: The outcome of interest, namely recorded pathology was fitted for association with 15 candidate SNPs found on chromosome 8q24. Four phenotypic variables (smoking status, alcohol consumption, age at diagnosis/joining the study, and body mass index (BMI)) were tested with the outcomes of interest using a multivariate analysis and squared data. Smoking status, alcohol consumption, and age at diagnosis (cases) were significantly associated with histology and hence these variables were adjusted for prior to further analyses. A generalised linear model was fitted to test the linearity of the genotype-phenotype relationship. For linearity of the genotype-phenotype relationship, an additive model was used whereby each SNP was coded 0, 1, and 2 (*Balding, 2006*).

All analyses were corrected for multiple testing using a false discovery rate (FDR) (*Storey, 2002*). SAS (V9.2 SAS Institute., Cary, NC, USA) and R (*R Foundation for Statistical Computing, 2012*) were used for statistical analyses.

## RESULTS

The control group comprised of healthy men with no history of cancer (excluding melanoma) and a PSA below 4 ng/ml. The control group was significantly younger (mean age = 53.1 years) than the men with prostate cancer (mean age 65.8 years). Within five years of enrolment 9 men from the control group had developed prostate cancer and were enrolled as volunteers on the Urology study. Tumour staging data were not used as less than 50% of the data were available. Gleason scores ranged from 4 to 10 with 36% of volunteers with a Gleason score of 6 or less, 45% with a Gleason score of 7 and 20% with a Gleason score of 8 or greater.

Smoking status was tested for association with prostate cancer (Table 2), and there were significantly more "Ever" smoked among those with prostate cancer compared to

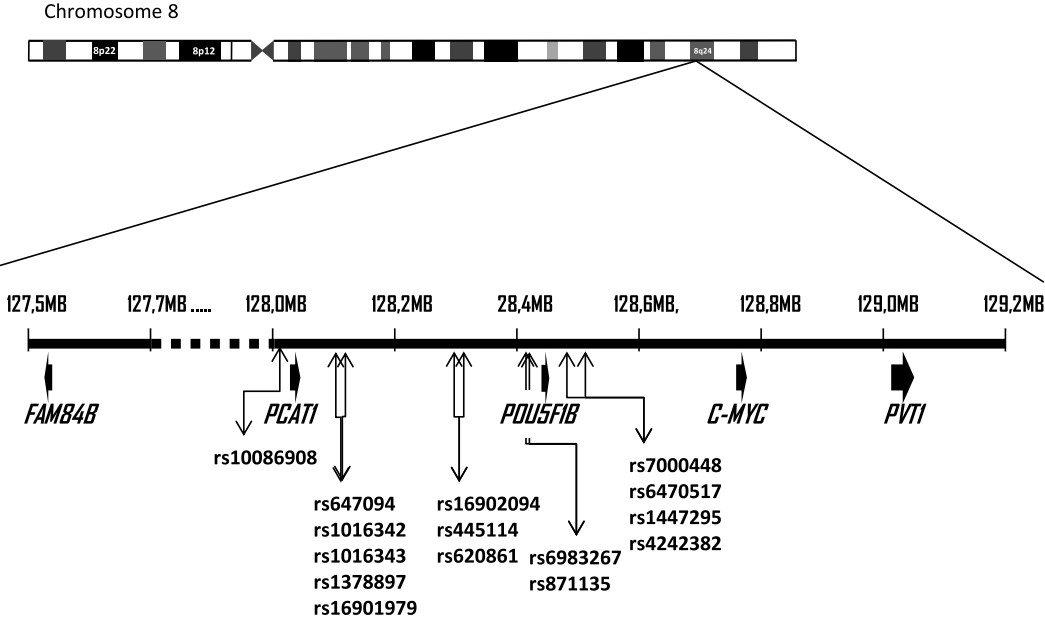

**Figure 1 Schematic of 8q24 chromosomal region.** Localisation of single nucleotide polymorphisms on 8q24.21 which were tested in this study and show association with prostate cancer risk and/or progression. *C-MYC* and *FAM84B* are known genes on 8q24 which border the so-called gene desert. The exact position of the SNPs can be found in Table 3. (Figure adapted from *Kastler et al., (2010)*).

controls, with an odds ratio (OR) of 2.28 (95% CI [1.62–3.21]). An OR of 0.57 (95% CI [0.37–0.90]) was associated with alcohol consumption, indicating that fewer men with prostate cancer drank one or more alcoholic units per week, than healthy men. After performing a multivariate analysis it was found that as healthy men age, they tend to drink less, whereas men with prostate cancer drink comparatively more than healthy men of a similar age ($p = 3.24$ e–07). In contrast, older men who smoked were more likely to have prostate cancer than those who never smoked ($p = 4.63$ e–14). No significant association was found between BMI and prostate cancer. Smoking status, alcohol consumption and age of diagnosis were adjusted for further genotypic analysis. SNP locations are shown in Fig. 1 and Table 3. A risk assessment on malignancy in association with the SNPs is shown in Table 4. Rs16901979 and rs6983561 were both found to be in strong LD ($D' = 0.999$, $p = 2.22$ e–16). For this reason rs6983561 was removed from all analyses. Four polymorphisms (rs10086908 (*T* allele), rs16901979 (*A* allele), rs1447295 (*A* allele) and rs4242682 (*A* allele)) showed an increased risk of prostate cancer compared to the controls before and after adjustment for confounding variables, and remained significant after applying an FDR (Table 4).

The wGRS15 and histology were significantly associated (Table 4) and graphically shown in Fig. 2. As the influence of the risk alleles on prostate cancer increases there is greater variance between the prostate cancer cases and healthy controls (Fig. 2).

**Table 3** Single nucleotide polymorphisms and known genes or pseudo genes in 8q24, listed by position on chromosome 8 (HapMap Genome Build 37.3).

| SNP/gene | Genotype | Chromosome Position | SNP to Chromosome | Sequence length |
|---|---|---|---|---|
| **FAM84B** | | 127564683–127570711 | | **6029** |
| rs10086908 | C/T | 128011937 | Fwd | |
| **PROSTATE CANCERT1** | | 128025399–128033259 | | **7861** |
| rs6470494 | C/T | 128087904 | Fwd | |
| rs1016342 | C/T | 128092455 | Fwd | |
| rs1016343 | C/T | 128093297 | Fwd | |
| rs1378897 | A/G | 128122659 | Fwd | |
| rs16901979 | A/C | 128124916 | Fwd | |
| rs16902094 | A/G | 128320346 | Fwd | |
| rs445114 | C/T | 128323181 | Fwd | |
| rs620861 | C/T | 128335673 | Rev | |
| rs6983267 | G/T | 128413305 | Fwd | |
| rs871135 | G/T | 128426393 | Fwd | |
| **POU5F1B** | | 128427857–128429455 | | **1599** |
| rs7000448 | C/T | 128441170 | Fwd | |
| rs6470517 | A/G | 128460404 | Fwd | |
| rs1447295 | A/C | 128485038 | Fwd | |
| rs4242382 | A/G | 128517573 | Fwd | |
| **MYC** | | 128748315–128753680 | | **5366** |
| **PVT1** | | 128902874–129113499 | | **210626** |

Notes.

SNP, Single nucleotide polymorphism; FAM84B, Family with sequence similarity 84, member B; PROSTATE CANCERT1, prostate cancer associated transcript 1; POU5F1B, POU class 5 homeobox 1B; MYC, v-myc avian myelocytomatosis viral oncogene homolog; PVT1, otherwise known as Pvt1 oncogene.

## DISCUSSION

In a Caucasian population residing in NZ, we evaluated clinical characteristics as well as 15 SNPs located in the 8q24 chromosomal region that may independently confer susceptibility to prostate cancer. The search for genetic and lifestyle risk factors for prostate cancer is challenging due to varying results from different populations throughout the world and the need to and difficulty associated with controlling for numerous variables. The results presented here help to answer the question regarding the impact of BMI, age, smoking and alcohol intake, as well as polymorphisms in 8q24, on the risk of developing prostate cancer.

Numerous genome wide association studies have been carried out in large case-control studies in order to identify SNPs with small effects on prostate cancer risk and progression. Over 80 common SNPs have been estimated to contribute to prostate cancer risk (*Han et al., 2015*) and 33% of familial risk is associated with known SNPs (*Al Olama et al., 2014*).

**Table 4   Genotype and prostate cancer risk.** Association of 15 prostate cancer susceptibility single nucleotide polymorphisms, found in 8q24, with risk of prostate cancer (after adjustment for age, alcohol consumption and smoking status).

| | | Tested Allele | N | OR (95% CI) | *p* value | *q* value |
|---|---|---|---|---|---|---|
| rs10086908 | Malignant | T | 364 | 1.64 (1.25–2.15) | **3.03E−04** | **0.0040** |
| | Control | | 390 | 1.00 | | |
| rs16901979 | Malignant | A | 368 | 2.58 (1.48–4.50) | **8.42E−04** | **0.0056** |
| | Control | | 388 | 1.00 | | |
| rs1447295 | Malignant | A | 362 | 1.70 (1.18–2.46) | **0.0047** | **0.0209** |
| | Control | | 389 | 1.00 | | |
| rs4242382 | Malignant | A | 362 | 1.60 (1.10–2.33) | **0.0133** | **0.0444** |
| | Control | | 388 | 1.00 | | |
| rs16902094 | Malignant | G | 366 | 1.31 (1.03–1.65) | 0.0260 | 0.0695 |
| | Control | | 387 | 1.00 | | |
| rs1016342 | Malignant | C | 364 | 1.41 (1.02–1.95) | 0.0369 | 0.0822 |
| | Control | | 389 | 1.00 | | |
| rs445114 | Malignant | T | 353 | 1.19 (0.93–1.54) | 0.1729 | 0.3300 |
| | Control | | 392 | 1.00 | | |
| rs620861 | Malignant | C | 370 | 1.14 (0.90–1.46) | 0.2798 | 0.3965 |
| | Control | | 369 | 1.00 | | |
| rs6983267 | Malignant | G | 362 | 1.14 (0.90–1.46) | 0.2803 | 0.3965 |
| | Control | | 391 | 1.00 | | |
| rs871135 | Malignant | G | 355 | 1.14 (0.89–1.44) | 0.2967 | 0.3965 |
| | Control | | 390 | 1.00 | | |
| rs7000448 | Malignant | T | 360 | 0.92 (0.72–1.18) | 0.5032 | 0.6113 |
| | Control | | 386 | 1.00 | | |
| rs1016343 | Malignant | T | 345 | 1.06 (0.79–1.41) | 0.7172 | 0.7986 |
| | Control | | 398 | 1.00 | | |
| rs6470494 | Malignant | T | 370 | 1.04 (0.81–1.34) | 0.7794 | 0.8004 |
| | Control | | 390 | 1.00 | | |
| rs1378897 | Malignant | A | 372 | 1.05 (0.64–1.73) | 0.8386 | 0.8004 |
| | Control | | 380 | 1.00 | | |
| rs6470517 | Malignant | A | 337 | 1.00 (0.68–1.45) | 0.9836 | 0.8762 |
| | Control | | 391 | 1.00 | | |
| wGRS15 | Malignant | | 283 | 1.10 (1.04–1.15) | **3.14E−04** | |
| | Control | | 341 | 1.00 | | |

**Notes.**

Bold text indicates statistically significant values.

N, Sample size; OR, Odds ratio; CI, Confidence interval; wGRS15, Weighted genetic risk score based on 15 single nucleotide polymorphisms; *Q*-value, *p*-value after correction for multiple testing (*Storey, 2002*).

Although in many cases the contribution to genetic risk and interaction with treatment type is understood, the interaction with environmental influences remains to be elucidated (*Eeles et al., 2014*).

One of the problems associated with comparing results of studies carried out in different countries (with differing PSA testing policies) and in different communities, with varying

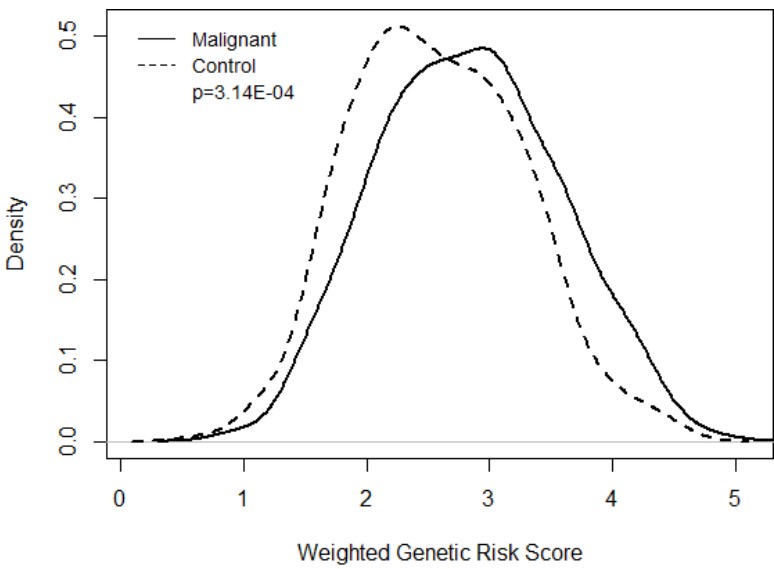

**Figure 2** **Genetic risk score.** Presentation of weighted genetic risk score differences using 15 genetic polymorphisms found in the 8q24 chromosomal region.

degrees of uptake of available medical services, is that men in the different studies are diagnosed at a different stage of prostate cancer. PSA is a biomarker that was initially used to monitor response to treatment and since the early 1990's has been used as an indicator for suspicion of asymptomatic prostate disease (*Cooner et al., 1990*). A problem arises as detection of prostate cancer using PSA as a screening method identifies lesions whose biological behaviour might not be the same as those identified due to clinical symptoms. In a country where a PSA screening programme is in place over a long period of time, one would expect most men to be recruited in the early stages of disease relative to a study carried out where PSA screening is not encouraged. Also, in the former one would expect to enrol more men with prostate cancer that might remain symptomless i.e., men who would die with prostate cancer and not from prostate cancer. NZ does not recommend PSA screening for asymptomatic men but the National Health Committee agree that prostate cancer is a suitable candidate for screening (*NHC, 2004*) and for example, one in four men over the age of 40 years living in the Waikato District of NZ were tested for PSA in 2010 (*Hodgson et al., 2012*). Of these, 71% were asymptomatic (*Hodgson et al., 2012*). For this reason one would expect a cohort of NZ men diagnosed with prostate cancer to consist of both early and later stage disease, and this is likely the case if Gleason score is considered as an indicator. Despite the increasing awareness of the risk of prostate cancer and the availability of the PSA test as an indicator of prostate cancer suspicion, those with a Gleason score of 8 or greater comprised 20% of the cohort, i.e., they were not diagnosed at an early stage of disease.

Numerous studies report no association between smoking and diagnosis of prostate cancer (*Giovannucci et al., 2007*; *Wolk, 2005*), although *Giovannucci et al. (2007)* and others have found that smoking is associated with an increased odds ratio of developing aggressive prostate cancer. In our study smoking tobacco was significantly associated with prostate

cancer risk (Table 2). No association was found between smoking and Gleason score (data not shown). Hereafter smoking was regarded as a confounding variable and adjusted for in further analyses (Table 2).

The lack of an association between BMI and prostate cancer incidence or more aggressive disease in our study could reflect the real lack of an association or it could be due to the inability of this study to adequately address this question. BMI was calculated at the time of enrolment, which varied from within 3 months to 8 years since diagnosis. However, just over 60% of volunteers with prostate cancer were enrolled within 12 months of diagnosis. BMI may have fluctuated over the lifespan and may also have changed since diagnosis and hence the measurement obtained may not reflect the BMI that influenced prostate cancer incidence. A cancer diagnosis sometimes evokes questions regarding causality, and some of our study participants self-reported a change in lifestyle with respect to smoking status, alcohol consumption, exercise and/or diet. Some of these lifestyle changes were maintained and were measurable (e.g., smoking cessation), whilst others (such as cooking methods and a decrease in animal fat intake) were not so easy to define and fluctuated over time. In addition to diagnosis, increasing age may influence lifestyle. In this study we know that fewer men with prostate cancer, drank alcohol than men who were healthy, but we also found that this association was confounded by age such that fewer older men with prostate cancer drank alcohol. Although there is evidence to support the view that alcohol consumption decreases with age, conflicting data also exists (*Thomas & Dilip, 1999*).

In contrast to the negative association between age and alcohol consumption, in this study age was positively associated with ever smoked and prostate cancer, thus supporting the view that risk of prostate cancer increases with age and smoking of tobacco. For this reason it is difficult to analyse the effect of some lifestyle changes on prostate cancer progression retrospectively and the time delay between diagnosis and enrolment is a limitation of this study.

Numerous risk loci for prostate cancer have been identified including a number in chromosome 8q24 (*Amundadottir et al., 2004*; *Helfand et al., 2010*; *Kim et al., 2010*; *Okobia et al., 2011*) and we tested 15 SNPs in chromosome 8q24.21 (Table 4) that were thought to contribute to prostate cancer risk, mortality, high Gleason scores or biochemical recurrence. The importance of this region was first implicated in prostate cancer in a genetic linkage analysis in Icelandic families (*Amundadottir et al., 2006*) and more recently has been proposed as a chromatin regulatory hub (*Du et al., 2015*). Chromosome aneusomy, including the gain of the 8q24 (*MYC*) region, has been reported from radical prostatectomy specimens (both carcinoma and adjacent tissues) in those with prostate cancer as well as from 15% of benign prostatic hyperplasia tissue (*Zhang et al., 2014*). This indicates that these chromosomal changes may occur in the progression of carcinogenesis in prostate tissue. Evidence for prostate cancer association within the region is particularly strong, with five distinct linkage disequilibrium (LD) blocks, spanning a 440-kb interval on 8q24 harbouring risk variants (*Al Olama et al., 2009a*; *Ghoussaini et al., 2008*). Although chromosome 8q24 is often described as a gene desert (*Kastler et al., 2010*; *Wasserman, Aneas & Nobrega, 2010*) there are a number of genes and SNPs in this region that may lead to the development of prostate cancer (Fig. 1) and could collectively play a role as a biomarker.

The closest known gene is *c-MYC* (*Hawksworth et al., 2010*). Other neighbouring genes include the pseudogene *POU5F1P1* (*Kastler et al., 2010*), *FAM84B* (*Ghoussaini et al., 2008*) and *PVT1* (*Meyer et al., 2011*) (Fig. 1).

*C-MYC* has an established role in carcinogenesis and is rearranged in approximately 15% of multiple myelomas (*Glitza et al., 2015*). *C-MYC* functions to regulate cell-cycle differentiation, proliferation and apoptosis (*Albihn, Johnsen & Arsenian Henriksson, 2010*) and hence it is an attractive candidate for an association with prostate cancer as these are important molecular targets for environmental variables in cancer prevention (*Davis & Milner, 2007*). *C-MYC* over-expression in prostate cancer enables androgen-independent growth and is associated with a Gleason score >5 (*Karan et al., 2002*; *Yang et al., 2005*). However, a number of authors have looked for an association between differences in *c-MYC* and miRNA transcription in normal and malignant tissues and 8q24 polymorphisms, yet no associations were found (*Pomerantz et al., 2009*; *Ribeiro et al., 2007*). Despite this, expression of *c-MYC* in early prostatic cancer tissues has been shown to be a good indicator for aggressive disease (*Hawksworth et al., 2010*).

The coding sequence for the gene *POU5F1B* is found in region 3 of chromosome 8q24 (also 4 exons and 3 transcriptional start sites are found in this region) and it is the only gene in this region with coding capacity. In addition it may play a role in regulating stem cell pluripotency (*Nichols et al., 1998*) and hence SNPs in this region may lead to prostate cancer susceptibility. Although *Kastler et al. (2010)* found a three-fold higher *POU5F1P1* expression in prostate cancer tissue versus surrounding normal tissue, and *Pal et al. (2009)* have found an association between SNP variants in this region and susceptibility to prostate cancer (and in the case of rs6470517 with aggressive prostate cancer), no association has consistently been made between gene expression and prostate cancer risk variants in 8q24 (*Gudmundsson et al., 2007*) or experimental design has not included both expression and risk variants (*Kastler et al., 2010*; *Pal et al., 2009*). SNPs that may be associated with *POU5F1P1*, namely rs1447295 and rs4242383, were found to be significantly associated with prostate cancer both before and after adjustment for confounders and after correction for multiple testing (Tables 2 and 4). Rs1447295 was one of the first variants in this region shown to have a strong association with prostate cancer risk in diverse populations (*Amundadottir et al., 2006*; *Robbins et al., 2007*; *Yeager et al., 2007*) and hence, in this respect, our data are consistent with that obtained by others.

Rs16901979 was found to be significantly associated with prostate cancer in this study). This is in general agreement with the article by *Gudmundsson et al. (2007)* where rs16901979 is believed to contribute 2–4% to the heritability of prostate cancer in Caucasian populations, but a vastly greater percentage of heritability in Negroid populations.

In addition to the difference in mean age between the cases and controls, a limitation of this study is the lack of a definitive test to confirm that those within the control group did not have prostate cancer. A negative biopsy of the prostate gland would be a suitable criterium for inclusion of volunteers into the control group, but this was not a reasonable expectation.

## CONCLUSION

Prostate cancer is one of the most common cancers amongst men, yet suitable biomarkers to accurately and reliably identify prostate cancer are not available. In this study both modifiable and non-modifiable risk factors for prostate cancer have been identified. A genetic risk score was calculated based on the 15 SNPs tested and found to be significantly associated with prostate cancer. Smoking and age significantly contributed to the risk of developing prostate cancer, and this risk was further increased by the presence of five SNPs in the 8q24 chromosomal region. The results presented here help to answer the question regarding the impact of modifiable risk factors, as well as polymorphisms in 8q24, on the risk of developing prostate cancer.

## ACKNOWLEDGEMENTS

The authors thank Tom Manly (AgResearch, NZ) and Philip Shepherd (The Liggins Institute, NZ) for running the Sequenom plates. The volunteers who made this study possible by consenting to participate in this study are acknowledged.

### Funding

This work was supported by funding received from the New Zealand Cancer Society, Auckland Cancer Society, New Zealand; The Goodfellow Trust, New Zealand; and The A+ Trust, Auckland District Health Board, New Zealand. The funders had no role in study design, data collection and analysis, decision to publish, or preparation of the manuscript.

### Grant Disclosures

The following grant information was disclosed by the authors:
New Zealand Cancer Society.
Auckland Cancer Society.
The Goodfellow Trust.
The A+ Trust, Auckland District Health Board.

### Competing Interests

Lynnette R. Ferguson is an Academic Editor for PeerJ.

### Author Contributions

- Karen S. Bishop conceived and designed the experiments, performed the experiments, wrote the paper, prepared figures and/or tables, reviewed drafts of the paper.
- Dug Yeo Han analyzed the data, contributed reagents/materials/analysis tools, wrote the paper, prepared figures and/or tables, reviewed drafts of the paper.
- Nishi Karunasinghe reviewed drafts of the paper, obtained ethical approval, set-up the cohorts; co-ordinated sample collection, processed the samples and stored them for future work.

- Megan Goudie reviewed drafts of the paper, recruited and enrolled the volunteers with prostate cancer, obtained blood samples and obtain clinical data.
- Jonathan G. Masters reviewed drafts of the paper, involved in patient recruitment, provided clinical advice.
- Lynnette R. Ferguson contributed reagents/materials/analysis tools, reviewed drafts of the paper.

## Human Ethics

The following information was supplied relating to ethical approvals (i.e., approving body and any reference numbers):

Health and Disabilities Ethics Committees: Northern Y Regional Ethics Committee, NZ. Ethics Ref: NTY/05/06/037 and NTY/06/07/060.

## Data Availability

Figshare: https://figshare.com/s/750929b0829d11e583c306ec4bbcf141.

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
