# Peer review of "An examination of clinical differences between carriers and non-carriers of chromosome 8q24 risk alleles in a New Zealand Caucasian population with prostate cancer"

_PeerJ, doi:10.7717/peerj.1731_

## Round 0.1 · original submission · Major Revisions

The reviewers have come up with some very salient concerns. In particular you are required to conduct a multivariate analysis to assess if smoking and alcohol are independent risk factors. Please also expand on the limitations

Reviewer 1 ·

Basic reporting

Prostate cancer is a major health problem, but the principal risk factors associated to the disease (age, ethnicity and family history ) can not be prevented. However, there must be other risk factors, such related to lifestyle, environment, etc. Great efforts are being made in an attempt to the decrease prostate cancer incidence. In this manuscript, the authors investigate the effect of smoking and alcohol intake as well as the impact of 16 SNPs in the 8q24region.

Although the results of the study could be relevant, a multivariate analysis is required to prove that smoking and alcohol intake are independent prognostic factors for prostate cancer in this series.

Experimental design

- A limitation of the study is the assumption that all controls are free from prostate cancer. That is not completely true since cancer investigations are only trigged by a PSA >4 and therefore, there could be some cases PSA<4 and undiagnosed prostate cancer. This should be acknowledged in the discussion.

- Table 2 reports smoking as current smoker/Ex-smoker/non-smoker, whilst in table 5 there are only 2 cathegories

Validity of the findings

Median age of cases is twelve years older than that of controls. Age is a well known risk factor for prostate cancer and authors should take this into consideration and stratify the analyses by age. They have found that smoking and alcohol intake are associated with prostate cancer risk, but may be than older patients drink and smoke more. A multivariate analysis including age, alcohol intake and smoking should be conducted before concluding that alcohol and smoking are independent prognostic factors for prostate cancer.

Additional comments

Minor comments:
- Variables should be defined in the Methods section. For example: Alchol consumption "NO", how much alcohol means: teetotal, once a week, not a drunk...

- "PSA levels were obtained at the time of prostate cancer diagnosis"..... should be removed from the Results section and included in the Methods section.

-Table 2 reports smoking as current smoker/Ex-smoker/non-smoker, whilst in table 5 there are only 2 categories: Ever smoked yes or not. For consistency, smoking always have the same categories.

-Table 2: Gleason score should not be reported in this table. Gleason categories are usually ≤6, 7, ≥8

-Table 5: Please, provide N for each category of smoking and alcohol consumption.

- The lack of association between BMI and prostate cancer should also be reported in the Results section.

- No mention is been made of the over 100 SNPs associated to prostate cancer risk. That should be added to the discussion to put your results in context.

Reviewer 2 ·

Basic reporting

The article i clear and well written.

Experimental design

A standard case control methodology has been used and the design is appropriate to the questions investigated.
The main concern with the study lies in the overall sample size for investigating the number of factors considered.

Validity of the findings

The findings are potentially informative and or illustrative of the risks seen in this particular population. No adjustment has been made for multiple testing which may result in potentially misleading results due to known and unknown confounding in other measured and unmeasured factors.

Additional comments

A nicely presented paper relevant to the population from which the sample has been derived. Quite a small sample size in comparison to other studies in this disease area these days where international pooling studies have resulted in investigations based on several tens of thousands of cases rather than hundreds but eh result still potentially have country specific value.

Reviewer 3 ·

Basic reporting

No Comments

Experimental design

No Comments

Validity of the findings

A major concern is that the phenotypic variables smoking status, alcohol consumption and BMI were collected post diagnosis. This was only mentioned in the discussion (should be described in the methods as well). As the authors allude to in the discussion, this can seriously bias their findings as the cases’ habits may have changed since diagnosis. As a result, I find it difficult to interpret the results.

Additional comments

Bishop et al review. I have a few comments that I hope the authors will find helpful.
Introduction
“Risk of prostate cancer is associated with both high penetrance prostate cancer susceptibility genes...”. This statement might be a bit of a stretch given that very few have been consistently replicated.
Methods
What were the response rates for the cases and controls?
A flow diagram showing the numbers would be helpful in describing the various exclusions.
“None of the SNPs at the time of selection were known to be in LD with each other”. Later in the discussion, the authors mention “Rs16901979 and rs6983561...are believed to be in strong LD”. SNPs in strong LD should be taken into account in the GRS.
A major concern is that the phenotypic variables smoking status, alcohol consumption and BMI were collected post diagnosis. This was only mentioned in the discussion (should be described in the methods as well). As the authors allude to in the discussion, this can seriously bias their findings as the cases’ habits may have changed since diagnosis. As a result, I find it difficult to interpret the results.
How do the authors define “alcohol consumption”? Does it seem peculiar that only 14% of controls have a “yes” status?
Results
“21 men from the control group had developed prostate cancer and were transferred to the “Malignant” group”. These men should be included both as cases and controls in the analysis.
“...there were significantly more current and ex-smokers among those with prostate cancer compared to controls”. That’s correct for current, but not for ex-smokers. CIs should be stated in the text for 5.83, 1.11.
“...there is greater variance between the prostate cancer cases and healthy controls”. Would prefer an alternative word to “variance”, as readers might get confused with statistical variance.
Discussion
I’m not sure what the paragraph about prostate cancer recurrence has to do with this study? I feel that it diverges somewhat from could be interpreted from the results.
Table 4
What is “Q-value”?
ORs could be reported to 2 decimal places.

---

## Round 0.2 · Major Revisions

You have responded well to the previous reviews but there are additional comments from the reviewers which require attention.

Reviewer 1 ·

Basic reporting

No Comments

Experimental design

Authors have conducted the analyses required by the reviewers.

Validity of the findings

Authors have acknowledged and explained the limitations of their findings in the discussion.

Additional comments

Just two minor comments:

1. Please, specify which is your age cut off when in the results section you describe that: “fewer older men with prostate cancer drank alcohol than younger”.
Is it men older and younger than the mean (65.8y), men in the first vs the forth quartile?......

2. Errata in line 183: It should say: Over 100 common SNPs (instead of 1000 SNPs) have been estimated…..

Reviewer 3 ·

Basic reporting

No Comments

Experimental design

No Comments

Validity of the findings

No Comments

Additional comments

Whilst some of the reviewers’ recommendations have been adequately addressed, there are still several outstanding issues.

Alcohol definition needs further clarification, “…consumed each week” over what period of time? In the last week? Last year? Lifetime?

ORs should have 95% CIs reported in the text

As for the issue about controls who become cases, you’ll need a better argument to convince me than just “The Statistician on the study feels that data cannot be used twice in the same analysis”. Read the section “control selection” in Rothman & Greenland’s Modern Epidemiology, p98 (2nd edition). The analysis should be redone, with controls who become cases be included as a case AND as a control.

“BMI may have changed since diagnosis”, not only since diagnosis, but it may also have changed during the latent period before clinical diagnosis.

With response rates, some mention should be made. I disagree that it is of low/no interest as it is an important issue; it indicates the representativeness (or lack of) of the entire population.

The proportion of missing data, particularly for the cases, is of concern. Have the authors considered multiple imputation?

Chromosome position, which build version is this based on?

---

## Round 0.3 · accepted · Accept

You have responded well to the reviewer comments